# Solvent-derived defects suppress adsorption in MOF-74

Yao Fu[1,2,3,4], Yifeng Yao[2], Alexander C. Forse [4,5,6], Jianhua Li[1], Kenji Mochizuki [2], Jeffrey R. Long [4,6,7], Jeffrey A. Reimer [4], Gaël De Paëpe [3] & Xueqian Kong [1,2,8] ✉

Defects in metal-organic frameworks (MOFs) have great impact on their nano-scale structure and physiochemical properties. However, isolated defects are easily concealed when the frameworks are interrogated by typical characterization methods. In this work, we unveil the presence of solvent-derived formate defects in MOF-74, an important class of MOFs with open metal sites. With multi-dimensional solid-state nuclear magnetic resonance (NMR) investigations, we uncover the ligand substitution role of formate and its chemical origin from decomposed N,N-dimethylformamide (DMF) solvent. The placement and coordination structure of formate defects are determined by $^{13}C$ NMR and density functional theory (DFT) calculations. The extra metal-oxygen bonds with formates partially eliminate open metal sites and lead to a quantitative decrease of $N_2$ and $CO_2$ adsorption with respect to the defect concentration. In-situ NMR analysis and molecular simulations of $CO_2$ dynamics elaborate the adsorption mechanisms in defective MOF-74. Our study establishes comprehensive strategies to search, elucidate and manipulate defects in MOFs.

Metal-organic frameworks (MOFs) are crystalline materials with diverse functionalities and tunable porosity[1,2]. While the crystalline matrix of MOFs enables the reticular design and precise control over their internal geometry and chemistry, the unavoidable presence of defects disrupts periodicity and alters significantly their local structures or properties[3–5]. In certain classes of MOFs, such as UiO-66, ZIF-8, HKUST-1 etc., the presence of defects has been frequently noticed, and the defect structure or density are sometimes engineered to optimize their properties[6–9]. Nevertheless, in thousands of other MOF materials, the defects remain elusive: neither the existence nor their structures are acknowledged.

MOF-74 or $M_2$(dobdc) ($H_4$dobdc = 2,5-dihydroxyterephthalic acid; M = Mg, Co, Ni, Zn, Mn, Fe) is a benchmark framework with outstanding adsorption properties for a range of applications, including hydrogen storage and hydrocarbon separations[10–13]. The structure of MOF-74 is generally described as hexagonal honeycomb channels with accessible open metal sites. Its favorable gas adsorption is attributed to the abundant under-coordinated metal sites after activation[14–16]. Some studies introduced defects into MOF-74 intentionally by modulating non-native organic ligands, which are typically called as engineered defects[17–20]. For example, Heidary and coworkers added a fraction of 2-hydroxyterephthalic acid in the synthesis of Ni-MOF-74 to modify the local Ni-O coordination bonds[17]. Villajos and coworkers partially mixed 2,5-dihydroxybenzoic acid and hydroxyterephthalic acid into Co-MOF-74 structure and increased the accessibility of open metal sites[18]. However, the existence of solvent-

---

[1]Department of Physical Medicine and Rehabilitation, Sir Run Run Shaw Hospital, Zhejiang University, Hangzhou 310027, PR China. [2]Key Laboratory of Excited-State Materials of Zhejiang Province, Department of Chemistry, Zhejiang University, Hangzhou 310027, PR China. [3]Univ. Grenoble Alpes, CEA, IRIG-MEM, Grenoble, France. [4]Department of Chemical and Biomolecular Engineering, University of California, Berkeley, CA 94720, USA. [5]Department of Chemistry, University of Cambridge, Cambridge CB2 1EW, UK. [6]Department of Chemistry, University of California, Berkeley, CA 94720, USA. [7]Materials Sciences Division, Lawrence Berkeley National Laboratory, Berkeley, CA 94720, USA. [8]Institute of Translational Medicine, Shanghai Jiao Tong University, Shanghai 200240, China. ✉e-mail: kxq@zju.edu.cn

derived defects (i.e., the defects related to the solvents used in MOF synthesis) in MOF-74 was largely unnoticed in the literature. Such solvent-derived defects could have unexpected impact on the adsorption or catalytic properties[21,22].

The characterizations of defects in MOFs and related porous materials have been a formidable challenge due to the insensitivity of techniques that assess local non-periodic structures. While correlated defects could be revealed by X-ray or electron diffraction[23–26], transmission electron microscopy[27–30], and vibrational spectroscopy[31–33], the structural determination of isolated defects is still difficult. We recently demonstrated that solid-state nuclear magnetic resonance (NMR), with the adaptation of advanced pulse sequences and multi-faceted strategies, can be powerful to identify not only the chemical environment of defects but also their geometries and distributions[34–36].

In the present work, we evidence the presence of solvent-derived formate defects in MOF-74, mainly thanks to the use of solid-state NMR. The formate present in the MOF-74 framework does not correspond to an intentionally introduced modulator but to a solvent byproduct (from N,N-dimethylformamide solvent) used in the common synthetic process of MOF (Fig. 1a)[10–12]. We first discover the existence of formate defects thanks to solid-state and solution-state NMR. Then we prove that formate moieties substitute the dobdc[4-] ligand in a 4:1 ratio, resulting in missing-linker defects. The structure of these defects is unraveled through two-dimensional $^{13}$C-$^{13}$C correlation solid-state NMR combined with density functional theory calculations. Interestingly, the concentration of defects can be controlled by varying the ratio of metal salt versus H$_4$dobdc in the precursor solution. We further show that the reduction in surface area and CO$_2$ uptake scale linearly with the formate concentration in defective MOF-74. The in situ NMR pattern analyses and molecular dynamics (MD)

simulations of CO$_2$ dynamics suggest that adsorption mechanisms in ideal and defective MOF-74 are similar except that the number of adsorption sites is reduced in defective MOF-74. Our study provides a detailed molecular picture of formate defects in MOF-74 and demonstrates their impact on adsorption properties.

## Results and discussion
### Identifying formate defects in MOF-74
Mg-MOF-74 is prepared according to the reported common procedure using Mg(NO$_3$)$_2$ and H$_4$dobdc in a solution mixed with N,N-dimethylformamide (DMF), ethanol and water[10,11]. The metal-to-ligand ratio is varied in the precursor solution to give different samples (Supplementary Table 1). Prior to characterizations, the solid products are washed with DMF and methanol. The formation of MOF-74 crystals is confirmed by powder X-ray diffraction (PXRD) (Fig. 1b and Supplementary Fig. 1). The set of MOF samples show similar PXRD patterns. Scanning electron microscope (SEM) images show that the MOF crystals form micron-sized rods as expected (Supplementary Fig. 2).

The $^{13}$C solid-state NMR spectra, however, reveal subtle differences (Fig. 1c). Besides the assigned resonances to the dobdc[4-] ligand, an unexpected $^{13}$C signal at -172 ppm is observed in most samples (Fig. 1c and Supplementary Fig. 3). Its distinct $^{13}$C chemical shift and the evidence of a direct CH bond are different from the carboxylate of dobdc[4-] or other existing chemicals in the precursor solution. In the $^1$H-$^{13}$C cross polarization (CP) spectra with a short contact time (CT = 50 μs), the emerging 172-ppm peak demonstrates a fast CP kinetics supporting its assignment to protonated carbon[37]. The possibility of a dangling -COOH is excluded, because the -COOH on H$_4$dobdc does not show a detectable signal under the same CP condition of 50 μs (Supplementary Fig. 4). In contrast, the non-protonated sites, e.g., the carboxylate, only

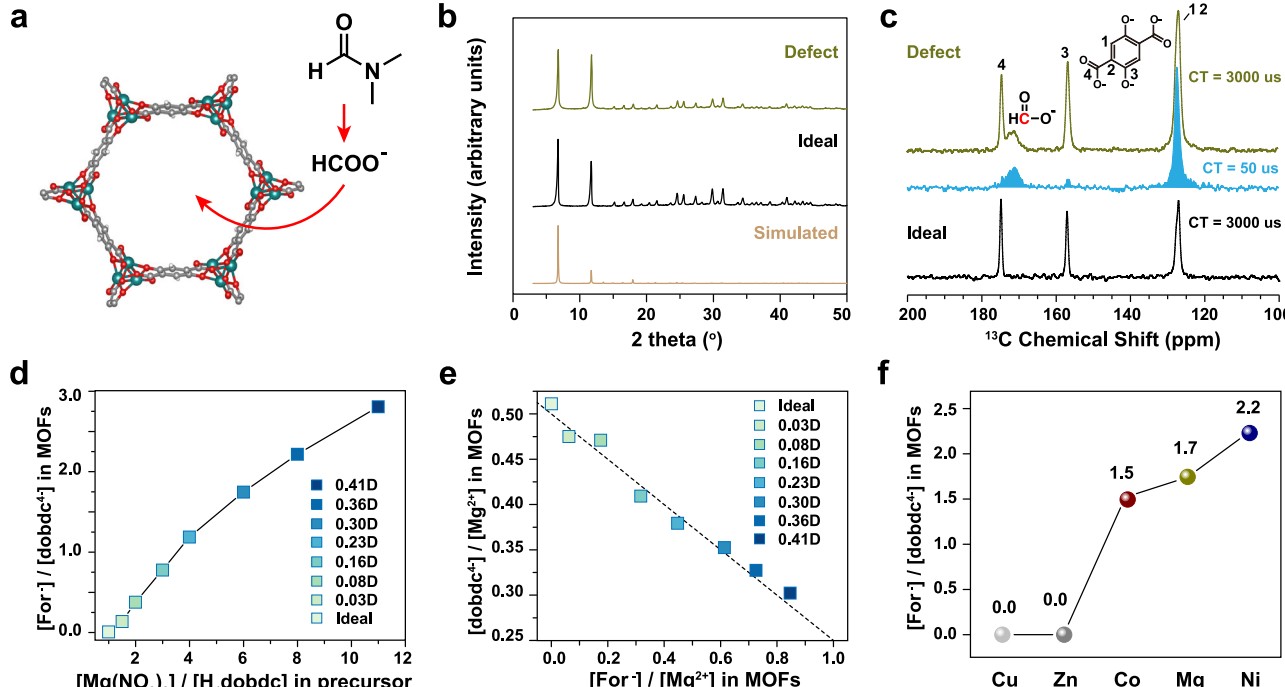

**Fig. 1 | The identification of formate defects in MOF-74. a** Schematic illustrations of the introduction of formate defects into MOF-74 from decomposed DMF. **b** PXRD patterns of the ideal sample and the defective 0.30D sample. **c** $^{13}$C CPMAS spectra of ideal and defective MOF-74 with a short contact time (CT = 50 μs, showing the carbons directly bonded by H) or with a long contact time (CT = 3000 μs, showing all carbon sites). **d** The concentration ratios of coordinated formate to dobdc[4-] in the MOF crystals ([For$^-$]/[dobdc[4-]]) plotted against concentration ratios of Mg(NO$_3$)$_2$ to H$_4$dobdc in the precursor solution ([Mg(NO$_3$)$_2$]/[H$_4$dobdc]). **e** The concentration ratios of dobdc[4-] to Mg$^{2+}$ ([dobdc[4-]]/[Mg$^{2+}$]) plotted against the concentration ratios of formate to Mg$^{2+}$ ([For$^-$]/[Mg$^{2+}$]) in MOFs. The dotted line indicates the charge balance: $(+2) \times [Mg^{2+}] + (-4) \times [dobdc^{4-}] + (-1) \times [For^-] = 0$. **f** The concentration ratios of coordinated formate to dobdc[4-] ([For$^-$]/[dobdc[4-]]) in MOF-74 of different metal ions, for the case of a metal-to-ligand concentration ratio of 6:1 in the precursor solution. Source data are provided as a Source Data file.

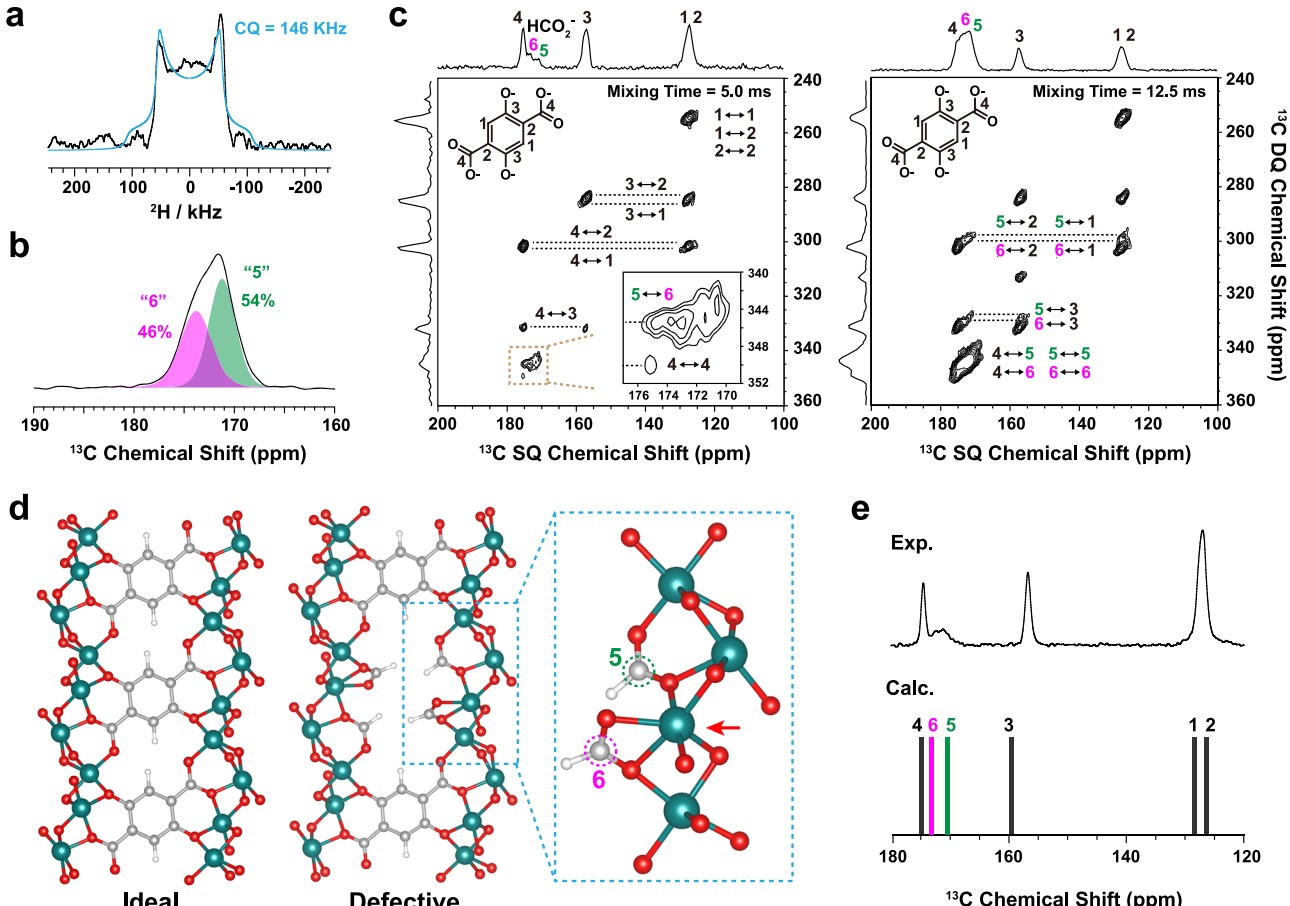

**Fig. 2 | The determination of formate coordination structure. a** Experimental (black) and fitted (blue) $^2H$ quadrupolar patterns of the $d_1$-formate in defective Mg-MOF-74. The NMR experiment was performed at 300 K. The quadrupolar splitting ($C_Q$) of $^2H$ pattern is -146 kHz. **b** The $^{13}C$ CPMAS spectrum of coordinated formate (CT = 50 μs) in defective Mg-MOF-74. The formate signal can be deconvoluted into two sites labeled as "5" and "6". **c** 2D $^{13}C$-$^{13}C$ double-quantum single-quantum (DQ-SQ) correlation spectra of sample 0.23D recorded at 100 K with different mixing times. **d** Ideal and defective Mg-MOF-74 structures optimized by DFT calculations. Blue, red, gray and white spheres represent Mg, O, C, and H atoms, respectively. Six-coordinated Mg is marked with an arrow. **e** Experimental $^{13}C$ CPMAS spectrum of defective Mg-MOF-74 and the calculated $^{13}C$ chemical shifts based on the defective structure in (**d**). Source data are provided as a Source Data file.

show up after a long contact time (CT = 3000 μs). This indicates the 172-ppm signal is a protonated carboxylate, i.e. formate, given its characteristic chemical shift and the strong coupling to proton[38]. The existence of formate is further confirmed by $^1H$ solution-state NMR of acid-digested Mg-MOF-74 sample, which shows the characteristic 8.2 ppm $^1H$ resonance for formic acid (Supplementary Fig. 5). Quantitative analysis of $^{13}C$ solid-state and $^1H$ solution-state NMR allows determining the formate to linker ratio in Mg-MOF-74 (Supplementary Fig. 6).

Interestingly, when the metal-to-ligand ratio (i.e. $[Mg(NO_3)_2]/[H_4dobdc]$) is 1:1 in the precursor solution, the $^{13}C$ and $^1H$ signals of formate is absent (Fig. 1c and Supplementary Fig. 5). This particular sample without formate is referred as "ideal" MOF-74 while others are regarded as "defective". The defective and ideal MOF-74 differ in their Brunauer-Emmett-Teller (BET) surface areas (Supplementary Fig. 7a). The BET surface area of Mg-MOF-74 decreases from ~1900 m²/g to ~700 m²/g as the defect concentration increases. Similarly, the MOF-74 samples of higher defect concentrations have lower $CO_2$ adsorption capacities (Supplementary Fig. 7b). This indicates the concentration of formate has a significant impact on gas adsorption capability, as we will discuss in the later section.

The concentration of formate in MOF-74 is related to the metal-to-ligand ratio used in the precursor solution (Fig. 1d). As the metal-to-ligand ratio is increased, the concentration of incorporated formate (measured as the formate-to-dobdc⁴⁻ ratio, $[For^-]/[dobdc^{4-}]$) increases. Inductively coupled plasma optical

emission spectrometry (ICP-OES) and quantitative solution-state NMR (Supplementary Table 2) show that metals and ligands in the crystals are in charge balance (dotted line in Fig. 1e): $(+2) \times [Mg^{2+}] + (-4) \times [dobdc^{4-}] + (-1) \times [HCO_2^-] = 0$. This determines the stoichiometry in the chemical formula of MOF-74: $Mg_2(dobdc^{4-})_{1-x}(HCO_2^-)_{4x}$. The defective samples are thus labeled as $x$D, where $x$ is considered as the defect concentration.

It is known that DMF can hydrolyze into formic acid in reactive conditions[39] and the degradation products of DMF have been observed in UiO-66 and other frameworks[25,40]. Here we show that different metals, including Co, Ni, Zn, and Cu, give rise to different effects on defect formation. When the samples are prepared with the same metal-to-ligand ratio of 6:1, Co-MOF-74 and Ni-MOF-74 (as well as Mg-MOF-74) exhibit high concentrations of formate defects. Yet Zn-MOF-74 and Cu-MOF-74 are almost free of formate defects (Fig. 1f and Supplementary Figs. 8–9).

It is worthwhile to prove that the formate incorporated into MOF-74 indeed originates from the DMF solvent. First, we used deuterated d₇-DMF as the solution and prepared defective MOF-74 (0.23D-$^2H$). $^2H$ solid-state NMR of the dried solid sample measured without magic-angle spinning (MAS) shows a broad $^2H$ quadrupolar pattern (Fig. 2a) consistent with a covalent C-D bond. The detection of $^2H$ signal validates that the decomposed product of DMF ends up in the MOF matrix. The broad $^2H$ quadrupolar pattern is consistent with a deuteron on carboxylate (C-D bond) rather than the deuterons on a methyl

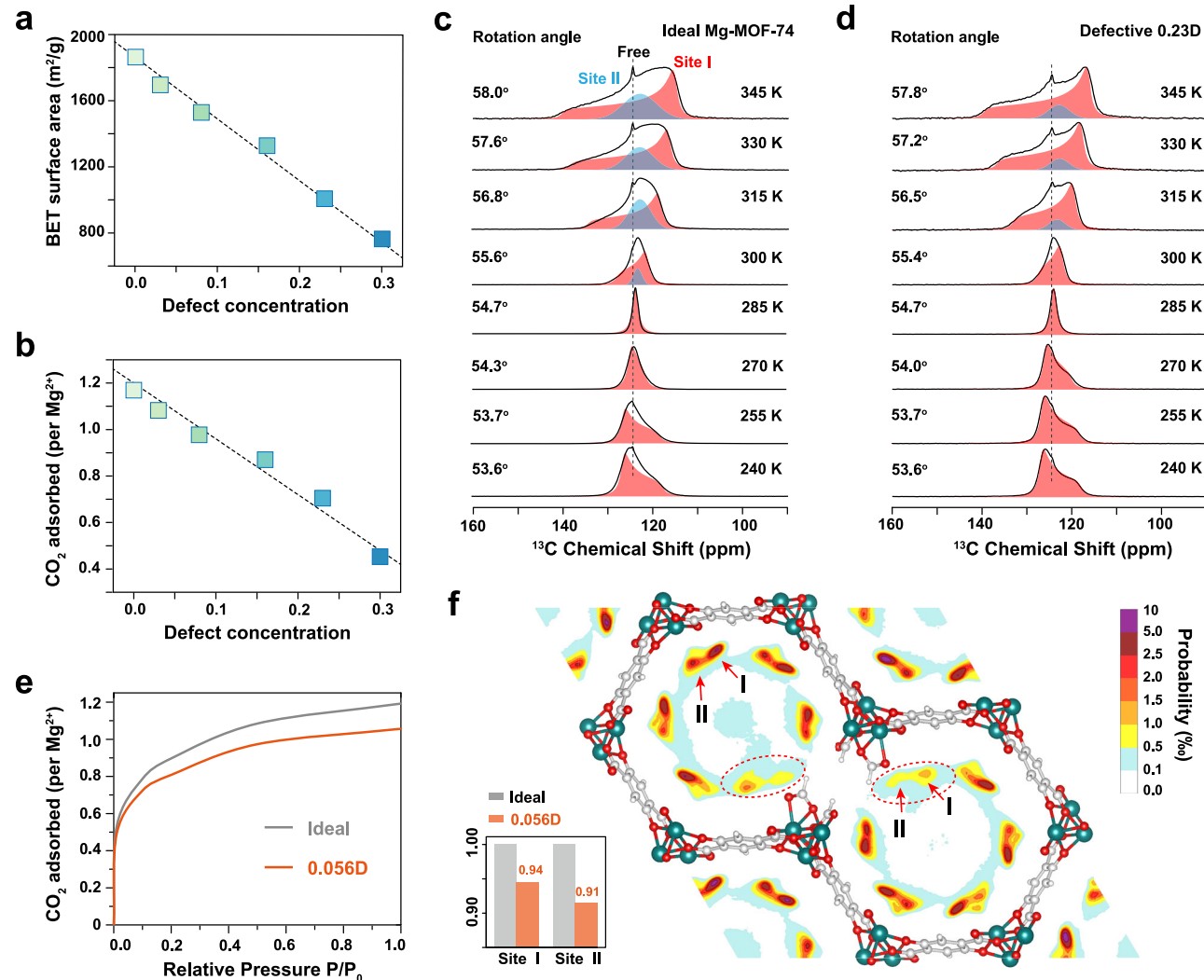

**Fig. 3 | The gas adsorption mechanisms in Mg-MOF-74. a** The BET surface areas of Mg-MOF-74 with various defect concentrations. The dashed line corresponds to the formula: $A_{BET} = 1860 \times (1-2x)\ [m^2/g]$ **b** $CO_2$ uptakes in Mg-MOF-74 at a pressure of 1 atm. The dashed line corresponds to the formula: $\frac{[CO_2]}{[Mg^{2+}]} = 1.2 \times (1-2x)\ [mol/mol]$. **c**, **d** $^{13}C$ CSA powder patterns of $CO_2$ in (**c**) ideal and (**d**) defective Mg-MOF-74 at variable temperatures. The red patterns are the simulated CSA lineshapes of $CO_2$ uniaxial rotation with corresponding rotational angles (attributed to the adsorption at site I). The blue patterns are Gaussian lineshapes for the $CO_2$ adsorption at site II. A vertical line is placed at 124.5 ppm to reference the isotropic chemical shift of free $CO_2$. **e** Simulated $CO_2$ adsorption isotherms in Mg-MOF-74 at 303 K by MD simulations. **f** Localization density of adsorbed $CO_2$ in the defective Mg-MOF-74. The site I and site II are marked with arrows. The dashed circles indicate the reduced localization density near the defect. The inserted plot in the lower left corner indicates the relative densities of $CO_2$ localization at site I and site II comparing defective and ideal frameworks. Source data are provided as a Source Data file.

group (i.e. $CD_3$), because the $^2H$ pattern of $CD_3$ would be much narrower due to methyl rotation[41]. Furthermore, when we synthesized 0.23D MOF sample with partial carbonyl-$^{13}C$ labeled DMF (4 and 22%), the 172-ppm $^{13}C$ peak becomes much more enhanced (Supplementary Fig. 10), confirming again the origin of formate from DMF. It is worth mentioning that, if the solvent is replaced with tetrahydrofuran (THF), the MOF-74 is free of formate defects under the same synthetic conditions (Supplementary Fig. 11).

## Determining the coordination structure of formate

We integrate multiple strategies to uncover the coordination structure of formate in the MOF. The first clue comes from the $^2H$ quadrupolar pattern of the deuterated formate (derived from $d_7$-DMF) thanks to its intrinsic sensitivity to molecular dynamics. The characteristic Pake pattern with a splitting of 146 kHz (i.e., the frequency difference between the tips) indicates the formate is completely rigid (Fig. 2a, Supplementary Fig. 12)[42], while a flexible formate would give a motional averaged pattern with narrowed splitting[43,44]. The rigidity of

formate implies that it should take the bidentate coordination instead of monodentate. Note that a monodentate coordination would result in wobbling motion similar to the acetate in UiO-66[34].

In Fig. 2b, the $^{13}C$ CPMAS spectrum (with a contact time of 50 μs) is shown for a defective MOF-74 with $^{13}C$ labeled formate (derived from 22% $^{13}C$-carbonyl labeled DMF). The use of a short contact time allows the selective observation of the formate and avoids the carboxylate signal from the dobdc$^{4-}$ ligands. The $^{13}C$ signal of formate is composed of two partially overlapped peaks with roughly equal populations (Fig. 2b): one at 171 ppm (labeled as "5") and the other at 174 ppm (labeled as "6"). These peaks indicate two distinct coordination structures of formate as we will discuss later. For instance, previous studies have shown that the chelating carboxylate normally has a larger $^{13}C$ chemical shift than the bridging carboxylate by about 3-5 ppm[45,46].

The next question regarding the defect structure is the spatial placement of formates in the framework, whether they are clustered together or distribute evenly with the dobdc$^{4-}$ ligands. We performed

$^{13}C$-$^{13}C$ double quantum-single quantum (DQ-SQ) experiments to explore the proximity between the organic components. The DQ-SQ spectra were recorded with MAS at 100 K to boost the signal-to-noise ratio of $^{13}C$ signals (Fig. 2c, Supplementary Fig. 13). The formate in the 0.23D sample is derived from 4% $^{13}C$-carbonyl labeled DMF. For a short recoupling time of 5 ms under the S3[47-49] dipolar recoupling sequence, the correlation peaks indicate short-range C–C proximity within 3–4 Å (Fig. 2c). The short-range correlation not only shows the intramolecular peaks for the dobdc$^{4-}$ ligand (marked as 1–2, 1–3, 1–4, 3–4, etc.), but also shows intermolecular correlation peaks (marked as 5–6) for the formate sites "5" and "6". This suggests the two formate configurations are positioned on the nearby Mg$^{2+}$ sites. As the recoupling time is increased to 12.5 ms, further C–C proximity within 6–7 Å can be revealed (Fig. 2c). The intermolecular correlations of 5–5 and 6–6 only become more pronounced at the long recoupling time, suggesting the 5–5 and 6–6 distances are farther than 5–6 distance (consistent with the proposed structure shown in Supplementary Fig. 14). Besides, the intermolecular correlations between the formate and the dobdc$^{4-}$ ligand are clearly observed including correlation peaks 5–4, 6–4, 5–3, 6–3, 5–1, 6–1 etc. The long-range peaks indicate that formate moieties are in proximity with dobdc$^{4-}$ linkers, and thus should distribute homogeneously in the framework.

With the experimental evidence, we conclude the formates can substitute dobdc$^{4-}$ ligands in Mg-MOF-74 and create defects on the wall of honeycomb channel. Considering four formate molecules substitute one dobdc$^{4-}$ to maintain charge balance, the bonding configurations of formates are optimized by density function theory (DFT) calculations (Fig. 2d). The calculated $^{13}C$ chemical shifts for formate sites "5" and "6" match with the experimental results (Fig. 2e). The simulated PXRD patterns of ideal and 0.056D defective frameworks are similar (Supplementary Fig. 15) suggesting the crystal lattice is unaffected by formate defects. However, the defective structure indicates that the formate defects reduce the number of open Mg$^{2+}$ sites in MOF-74. The Mg$^{2+}$ site shared by two formates is six-coordinated (marked with a red arrow in Fig. 2d), yet the Mg$^{2+}$ sites in an ideal framework are all five-coordinated. Therefore, a single defect site (where a dobdc$^{4-}$ linker is replaced by four formates) eliminates two open metal sites. We expect these formate defects should have notable impact on the adsorption properties of MOF-74.

## Understanding the impact of defects on gas adsorption

The BET surface areas of defective Mg-MOF-74 are measured with $N_2$ adsorption isotherms at 77 K (Supplementary Fig. 7a). The surface areas show a linear dependence with respect to the defect concentration: $A_{BET} = 1863 \times (1 - 2x) \, [m^2/g]$ (Fig. 3a). The $CO_2$ uptakes of Mg-MOF-74 at 303 K are plotted in Fig. 3b at a pressure of 1 atm. At 303 K, the molar uptake of $CO_2$ is about 1.2 : 1 with respect to the Mg sites in ideal Mg-MOF-74. The $CO_2$ uptake also shows a linear dependence with respect to the defect concentration: $\frac{[CO_2]}{[Mg^{2+}]} = 1.2 \times (1 - 2x) \, [mol/mol]$. These results indicate that a missing dobdc$^{4-}$ linker eliminates two adsorption sites for both $N_2$ and $CO_2$, and the observations are consistent with the two saturated Mg on a defect site in Fig. 2d. The reduction of open metal sites also reduces the heat of adsorption for $CO_2$ in defective Mg-MOF-74 (Supplementary Fig. 7e).

To further understand the mechanism of $CO_2$ adsorption, we perform in situ $^{13}C$ NMR of $CO_2$ adsorbed in Mg-MOF-74 (Fig. 3c, d). The $CO_2$ molecules are 99% $^{13}C$ labeled and are maintained at a pressure of 2 atm. The chemical shift anisotropy (CSA) patterns of $CO_2$ are collected at variable temperatures from 240 to 345 K. The dynamics of adsorbed $CO_2$ is inferred in the motional averaged CSA patterns[14,44]. The $CO_2$ in Mg-MOF-74 displays a dominant CSA pattern of uniaxial rotation (red pattern) which is interpreted as a multisite hopping along the channels[50]. The rotation angle varies with respect to the

temperature and the transitions are similar for both ideal and defective MOFs (Figs. 3c, 3d and Supplementary Fig. 16). The results indicate that the electrostatic interaction between $CO_2$ and the open metal sites is the primary (site I) adsorption mechanism for both ideal and defective MOFs[14,21,51-54]. The difference appears at temperatures above 300 K. Here, an additional Gaussian peak (blue pattern) appears in the ideal MOF-74, but it is much less evident in defective MOFs. The Gaussian peak in ideal MOF-74 is attributed to the second $CO_2$ adsorption sites (site II) in addition to the open Mg$^{2+}$ centers[14,22,52].

To further explore our experimental findings, we perform molecular dynamics (MD) simulations of $CO_2$ in Mg-MOF-74. The simulated $CO_2$ adsorption isotherms (at 303 K) for ideal and defective Mg-MOF-74 are shown in Fig. 3e. The $CO_2$ uptake for defective MOF (0.056D) is just 90% of that in ideal MOF at different partial pressures. This agrees well with the $1 - 2x$ formula mentioned above. In Fig. 3f, we plot localization density map of $CO_2$ adsorbed in the defective framework. As we can see, the $CO_2$ molecules are preferentially localized next to the Mg$^{2+}$ metal ions with site I and site II. However, at the formate defect, the localization density of $CO_2$ is greatly reduced for both primary I and secondary II sites.

Figure 3f clearly demonstrates that the reduced $CO_2$ uptake in defective Mg-MOF-74 is due to the local elimination of open metal sites by the formates (see the dashed circles). But the remaining intact Mg$^{2+}$ sites can still interact with $CO_2$ as strongly as those in an ideal framework. As a result, $CO_2$ molecules in the defective framework also hop along the honeycomb channel giving similar $^{13}C$ CSA patterns of uniaxial rotation. Furthermore, when the localization densities of $CO_2$ at site I and site II are counted separately (the inserted plot in Fig. 3f), it shows the adsorption at site II is more reduced than site I in the defective framework. This explains the apparent reduction of the Gaussian peak ($CO_2$ adsorption at site II) in the defective MOFs such as 0.23D (Fig. 3d) and 0.08D (Supplementary Fig. 16).

In summary, we uncover the solvent-derived formate defect in MOF-74 frameworks based on comprehensive NMR signatures. We show that formate is derived from decomposed DMF solvent, and its concentration can be controlled by the metal-to-ligand ratio in the precursor solution. The formate substitutes the dobdc$^{4-}$ ligand quantitatively and forms extra Mg-O bonds. The open metal site at the defect is eliminated and it results in the reduction of gas adsorption in defective MOF-74. Besides $CO_2$, the adsorption of hydrocarbons in MOF-74 could also be reduced because they are attracted to the open metal sites[55]. On the other hand, the formate defects alter the channel connectivity of MOF-74. It offers a possibility to control the diffusion pathway for small molecules, similar to the correlated defects in UiO-66 (also formed with formate)[23,24]. Remarkably, we show that the formate in MOF-74 can withstand temperatures up to 523 K and the defective MOF retains crystallinity after the formate is removed at 573 K (Supplementary Fig. 17). The heating treatment or other post-synthetic modifications could provide potential routes to engineer new functions in defective MOF-74[7,56,57].

## Methods
### Synthesis of Mg-MOF-74
The Mg-MOF-74 samples were synthesized by modifying a recipe from a previous report[10]. The ligand H$_4$dobdc (0.111 g, 0.559 mmol, 1 equiv.) and salt Mg(NO$_3$)$_2$·6H$_2$O (1 to 11 equiv., quantities given in Supplementary Table 1) were dissolved in 100 mL reaction glass bottles containing 50 mL of a 15:1:1 (v/v/v) mixture of DMF/ethanol/water solution by ultrasonication. Each reaction bottle was capped tightly and placed in an oven at 125 °C for 20 h. The solid products were washed with DMF 2 times every 3 h and with methanol 4 times every 12 h. For further activation, selected samples were activated at 523 K (to remove the solvent) or 573 K (to remove the solvents and formate) under vacuum for 1 day.

100% carbonyl-$^{13}$C labeled DMF was diluted with non-labeled DMF to 4 and 22% carbonyl-$^{13}$C labeled DMF. For $^{13}$C-labeled or $^2$H solid-state NMR experiments, three samples were synthesized by adding Mg(NO$_3$)$_2$·6H$_2$O (0.1115 g, 0.447 mmol, 4 equiv.) and H$_4$dobdc (0.022 g, 0.112 mmol, 1 equiv.) in 20 mL vials containing either 10 mL 4% carbonyl-$^{13}$C labeled DMF, 10 mL 22% carbonyl-$^{13}$C labeled DMF or 10 mL d$_7$-DMF. 0.67 mL ethanol and 0.67 mL water were also added in each vial. Each vial was capped tightly and placed in an oven at 125 °C for 20 h. The solid products were washed with DMF 2 times every 3 h and with methanol 4 times every 12 h. The solids were stored in a vial of methanol until they were used for further experiments.

Mg-MOF-74 samples were also synthesized in tetrahydrofuran (THF) solvent[58]. 0.75 mmol of H$_4$dobdc was dissolved in 10 mL of THF in a 30 mL vial. An aqueous sodium hydroxide solution (3 mL, 1 M) was added to this solution. Mg(NO$_3$)$_2$· 6H$_2$O aqueous solution (5 mL, 0.3 M) was then added to the vial. This vial was capped tightly and placed in an oven at 110 °C for 3 days. The resulting light-yellow powder was filtered, repeatedly washed with THF, and dried at room temperature.

### Synthesis of Co-MOF-74
A solid mixture of H$_4$dobdc (0.042 g, 0.213 mmol, 1 equiv.) and Co(NO$_3$)$_2$·6H$_2$O (0.371 g, 1.275 mmol, 6 equiv.) was added a 1:1:1 (v/v/v) mixture of DMF/ethanol/water (18 mL) in a 30 mL vial[10]. The suspension was mixed and ultrasonicated. The reaction vial was capped tightly and placed in an oven at 100 °C for 24 h. The solid products were washed with DMF 2 times every 3 h and with methanol 4 times every 12 h.

### Synthesis of Ni-MOF-74
A solid mixture of H$_4$dobdc (0.042 g, 0.213 mmol, 1 equiv.) and Ni(NO$_3$)$_2$·6H$_2$O (0.371 g, 1.275 mmol, 6 equiv.) was added a 1:1:1 (v/v/v) mixture of DMF/ethanol/water (18 mL) in a 30 mL vial[10]. The suspension was mixed and ultrasonicated. The reaction vial was capped tightly and placed in an oven at 100 °C for 24 h. The solid products were washed with DMF 2 times every 3 h and with methanol 4 times every 12 h.

### Synthesis of Cu-MOF-74
A solid mixture of H$_4$dobdc (0.042 g, 0.213 mmol, 1 equiv.) and Cu(NO$_3$)$_2$·3H$_2$O (0.308 g, 1.275 mmol, 6 equiv.) was added a 20:1 (v/v) mixture of DMF and 2-propanol (18 mL) in a 30 mL vial[59]. The suspension was mixed and ultrasonicated. The reaction vial was capped tightly and placed in an oven at 100 °C for 20 h. The solid products were washed with DMF 2 times every 3 h and with methanol 4 times every 12 h.

### Synthesis of Zn-MOF-74
A solid mixture of H$_4$dobdc (0.042 g, 0.213 mmol, 1 equiv.) and Zn(NO$_3$)$_2$·6H$_2$O (0.379 g, 1.275 mmol, 6 equiv.) was added a 20:1 (v/v) mixture of DMF and H$_2$O (18 mL) in a 30 mL vial[60]. The suspension was mixed and ultrasonicated. The reaction vial was capped tightly and placed in an oven at 100 °C for 20 h. The solid products were washed with DMF 2 times every 3 h and with methanol 4 times every 12 h.

### Characterizations
$^1$H solution-state NMR spectra were acquired on Bruker AV-300 or ABV-400. ~5 mg of each MOF-74 powder was digested in a solution of 0.15 mL of 35 wt.% DCl in D$_2$O and 0.4 mL of DMSO-d6 to quantify the formate and dobdc$^{4-}$ concentrations. The solutions were sonicated until the solids were fully dissolved. The actual formate and dobdc$^{4-}$ concentrations in MOF samples were determined from the $^1$H signals of formate and dobdc$^{4-}$ relative to that of known-concentration of 1,3,5-trimethoxybenzene (as the internal standard for quantification).

PXRD measurements were carried out on samples placed on a quartz holder using a Rigaku Ultimate-IV X-ray diffractometer

operated at 40 kV/30 mA with Cu Kα line ($\lambda = 1.5418$ Å). Patterns were collected in reflectance Bragg-Brentano geometry in the 2θ range from 3 to 50°.

SEM imaging was performed on a HITACHI SU8000 FE-SEM with a field emission at 5 kV.

Nitrogen sorption was measured at 77 K on BelSorp-max instrument. Brunauer-Emmett-Teller (BET) surface areas were calculated by fitting the isotherm data in the P/P$_0$ range of 0–0.1.

Carbon dioxide sorption was measured at 303 and 323 K. Prior to adsorption measurements, the Mg-MOF-74 samples were pretreated (activated) under vacuum for 24 h at 523 K.

ICP-OES analysis was conducted on an ICP Optima 7000 DV instrument. ~5 mg of MOF were sonicated and fully digested in 0.15 mL of 35 wt.% HCl solution and 0.4 mL of DMSO. 50 μL dissolved MOF solutions were diluted with 10 mL pure H$_2$O. The diluted solutions were used to test the magnesium contents by ICP-OES. The Mg$^{2+}$ concentration (mg·L$^{-1}$) in different MOF samples was experimentally determined by referring to a magnesium standard curve.

### Solid-state NMR
Room temperature $^{13}$C NMR experiments and variable-temperature static $^2$H NMR experiments were performed on a Bruker Avance III HD 400 MHz NMR spectrometer ($^1$H, 400.13 MHz; $^{13}$C, 100.61 MHz; $^2$H, 61.42 MHz) using a 3.2 mm magic angle spinning (MAS) probe. $^{13}$C spectra were collected using cross-polarization (CP) or direct polarization (DP) sequences under MAS of 15 kHz. The recycle delay of DP sequence was set to 250 s to ensure all the $^{13}$C signals were recovered to equilibrium, while the recycle delay of CP was set to 1.5 s. The $^1$H radio frequency (RF) field strength was 100 kHz and the $^{13}$C RF field strength was 83 kHz. The $^{13}$C signals are referenced to the methylene signal of adamantane at 38.5 ppm.

The static $^2$H solid-state NMR spectra were obtained by the solid echo sequence (90°$_x$-τ$_1$-90°$_y$-τ$_2$-acquisition). τ$_1$ was set to 100 μs and τ$_2$ was set to 0 to obtain the complete echo signal. Scans of 70 k were accumulated for each spectrum. The temperature was controlled by a Bruker temperature controller BCUII with a deviation of less than ±1.0 K. $^2$H line shape fittings were carried out by TOPSPIN.

$^{13}$C–$^{13}$C double quantum-single quantum (DQ-SQ) experiments were performed on a Bruker Avance III 400 MHz system equipped a low temperature (-100 K) double resonance 3.2 mm MAS probe. The 2D spectra were recorded at 100 K and a MAS rate of 12 kHz. Dipolar recoupling sequence S3[47] was used for DQ excitation and reconversion. 100 kHz RF-field strength was used for SW$_f$-TPPM[61] heteronuclear decoupling during indirect (t$_1$) and direct (t$_2$) detection periods, and for continuous wave (CW) decoupling during S3 recoupling. A z-filter of 100 μs was inserted before acquisition. Five experiments were recorded with different mixing times ranging from 1 to 5 loops of S3 for both the DQ excitation and reconversion blocks.

The variable-temperature in situ $^{13}$C NMR experiments were performed on a Bruker Avance III HD 400 MHz NMR spectrometer using a static probe. The activated samples were packed in a homebuilt sample tube and kept under a constant pressure of $^{13}$C-enriched CO$_2$ (2 atm) by connecting to a gas cylinder. The temperature was controlled by a Bruker temperature controller BCUII with a deviation of less than ±1.0 K. Single pulse excitation with a recycle delay of 5 s was used. Motional averaged NMR line shapes were simulated by EXPRESS package[62]. A linear geometry of CO$_2$ molecules with CSA tensors $\delta_{11} = \delta_{22} = 232$ ppm, $\delta_{33} = -82.5$ ppm was assumed. The simulation was performed under the fast motion limit (i.e., assuming the rotation rate exceeds $1 \times 10^6$ Hz) with 3-site jumps of equal populations (i.e., a rotation with 3 discrete steps).

### Computational methods
**System and force field.** The structural optimization and partial charge calculations were performed in CP2K package[63], whose input files were

generated by Multiwfn package[64]. The trigonal unit cell of MOF-74 is taken from previous work[22]. The atomic positions and cell size are optimized by the density functional theory (DFT) calculations, in which we use Perdew-Burke-Ernzerhof (PBE) exchange correlation functional[65] in combination with molecularly optimized (MOLOPT) Double-Zeta-Valence plus Polarization (DZVP)[66] and Goedecker-Teter-Hutter (GTH) pseudo potentials[67] basis set. The plane-wave energy cutoff is 400 Ry. The integration over the irreducible Brillouin zone is computed over a $2 \times 2 \times 8$ Monkhorst-Pack mesh of $k$-points. Then, the duplicated unit cell in the (001) direction is used as the intact MOF-74, which is expressed as $Mg_{36}(dobdc)_{18}$.

The defective MOF-74, $Mg_{36}(dobdc)_{17}(HCO_2)_4$, is built by replacing one dobdc$^{4-}$ linker in the intact MOF-74 by four formate groups, in which the defect concentration $x$ is 0.056 in the formula of $Mg_2(dobdc)_{1-x}(HCO2)_{4x}$. The atomic positions and cell size are optimized in the same way for the intact MOF-74, except the calculation being only performed on Gamma point due to its duplicated size.

The partial charges for both the intact and defective MOF-74 are calculated by the REPEAT scheme[68]. For the defective MOF-74, we consider that twelve Mg ions next to the formate molecules have an identical charge and two formate molecules on the diagonal have the same charge set.

For the 6–12 Lennard-Jones (LJ) potentials, Universal Force Field (UFF)[69] is applied to the MOF, while TraPPE model[70] is for $CO_2$. The cross interactions involving $CO_2$ are described by the model 3 in the paper by Lin et al.[71], while the Lorentz-Berthelot combination rules are applied for the others. The carbon and oxygen atoms of formate are considered as $C_a$ and $O_a$ in ref. 71, respectively, and the model 3 cross interaction is used. The intermolecular interactions are truncated at 1.2 nm, and the Particle-Particle Particle-Mesh K-space (PPPM) technique[72] with k-space accuracy of $1.0 \times 10^{-4}$ is used to compute the long-range Coulombic interaction.

**NVT-MD/GCMC simulations.** All the simulations were conducted in LAMMPS package[73]. The adsorption of $CO_2$ on the MOF is evaluated through the combination between MD simulations under the constant volume and pressure ensemble (NVT-MD) and the grand canonical Monte Carlo (GCMC) simulations. The NVT-MD/GCMC is a hybrid modeling approach that uses MD to advance atomic trajectories and GCMC to implement atomic insertions and deletions. The time step of MD simulations is 2 fs. GCMC exchanges $CO_2$ in the MOF with an imaginary ideal gas reservoir with a specified chemical potential (pressure). Each five trials for insertions and deletions are conducted on every 2 ps of MD simulation. The temperature is set to 303 K, which is maintained by Nose-Hoover thermostat[74,75]. The total simulation length at each pressure is 6 ns, and the last 2 ns is used for analyses. The calculated adsorption isotherms are scaled down to 70% to match the experimental results.

**$^{13}$C chemical shift calculations.** Gauge-invariant atomic orbital (GIAO) scheme[76–79] was applied to compute the isotropic magnetic shielding ($\sigma_{calc.}^{X}$) of $^{13}$C in the optimized unit cell of defective MOF-74 with the periodic boundary condition. The calculation was performed under the level of revTPSS/pcsSeg-1[80–82]. The individual carbon chemical shift of defective MOF-74 ($\delta_{calc.}^{X}$) is computed according to the equation: $\delta_{calc.}^{X} = \sigma_{calc.}^{ref.} + \sigma_{calc.}^{X} + \delta_{calc.}^{ref.}$. Here, methylene of adamantane ($\delta_{calc.}^{ref.} = 38.5$ ppm) is referenced. The calculated $^{13}$C chemical shifts are all shifted to downfield by 8 ppm to match the experimental results.

**Localization density map in Fig. 3f.** As mentioned above, the system size of the defective MOF-74 is double of the unit cell along the (001) direction. The defects (four formates) are introduced into the top half. We compute the local probability of carbon atom in the top half region and project them on the two-dimensional map perpendicular to the (001) direction.

**Input files for NVT-MD/GCMC simulations and NMR analyses.** The input files for NVT-MD/GCMC simulations and NMR calculations are available in Source Data file. Details are described in the file "File_List.pdf".

## Data availability

All data generated in this study are provided in the article and Supplementary Information, and raw data are provided in the Source Data file. Source Data file also contains the structural file for the defective MOF and input files for molecular dynamics simulation and $^{13}$C chemical shift calculations. Source data are provided with this paper.

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

## Acknowledgements

This work is financially supported by National Natural Science Foundation of China (21922410, 22072133, 22275159, X.K.); Zhejiang Provincial Natural Science Foundation (LR19B050001, X.K.); Leading Innovation and Entrepreneurship Team of Zhejiang Province (2020R01003, X.K.); CEA ECC program (Programme Economie Circulaire du Carbone-Projet HyperCool NMR, G.D.P.), the French National Research Agency (CBH-EUR-GS and ARCANE ANR-17-EURE-0003, G.D.P.), the "Investissements d'avenir" program (ANR-15-IDEX-02, CDP-DefiCO2, G.D.P.) and the European Research Council Grant ERC-CoG-2015 (No. 682895, G.D.P.). US Department of Energy (DoE), Office of Science, Office of Basic Energy Sciences under award number (DE-SC0019992, J.R.L. and J.A.R.).

## Author contributions

Conceptualization: Y.F., X.K., J.A.R., A.F., and G.D.P.; Sample preparations: Y.F.; Characterizations and solid-state NMR: Y.F.; Computational methods: Y.Y. and K.M.; Data analysis and writing: Y.F., X.K., J.A.R., A.F., G.D.P., J.R.L., Y.Y., K.M., and J.L.

## Competing interests

The authors declare no competing interests.
