## [Peer Review File · Nature Communications]

Solvent-derived Defects Suppress Adsorption in MOF-74REVIEWER COMMENTS

Reviewer #1 (Remarks to the Author):

The manuscript from Fu et al. describes an in-depth investigation of defects in MOF-74 resulting from the decomposition of DMF to formate. The manuscript is succinct and portrays a comprehensive analysis using a combination of multi-dimensional solid-state nuclear magnetic resonance and computational chemistry simulations to discover the effect of ligand substitution with formate and its chemical origin. This is impressive work and outlines how to discover defects for a broad readership, which effectively advertises the transferability of this approach.

I am pleased to find the simulations, used to confirm the experimental results, employed the freely available and established codes cp2k and lammps. I urge the authors to share the simulation structures, input files and raw data for this study, so that the community is able to reuse these structures identified and build upon this study. For example, models of the defective Mg-MOF-74 structures should be deposited so that other researchers can further investigate these materials.

I would like to commend the authors on this impressive study. The identification of DMF byproducts that form defect sites is an important discovery. This work represents reliable science with a wealth of data and information that is of the quality and novelty expected for publication in Nature Communications.

Reviewer #2 (Remarks to the Author):

Y. Fu and co-workers apply a range of solid state NMR techniques to investigate the concentration, localization, and ramifications of formate-based defects in MOF-74(Mg). The results are complemented by various other experimental techniques such as PXRD, N₂ and CO₂ sorption experiments and MD simulations. The occurrence of defects in MOFs and their impact on the properties, in this case on the gas adsorption mechanism, is an important and likewise challenging area and I rate the general topic as suitable for Nature Communications.

The strong part of the paper is certainly the results coming from the various applied solid state NMR methods. In the most general terms, the results confirm of what can be expected based on chemical intuition. For instance, it is shown that one dobdc^{4-} ligand leaves four positive charges unbalanced, and accordingly there is a 4:1 ratio between dobdc^{4-} defects and incorporated formate ligands, i.e. the observed stoichiometry is $\text{Mg}_2(\text{dobdc}^{4-})_{1-x}(\text{HCO}_2^-)_{4x}$. Solid state NMR provides additional spatial information, showing that the substitution pattern as shown in Figure S14 is indeed the most likely scenario. Moreover, the results indicate the absence of any strong correlated defects which have previously been found in other defective MOFs. Lastly, in situ ¹³C NMR of CO₂ adsorbed in the MOF was performed and complemented with MD simulations to show the elimination of open metal sites as the origin for reduced N₂ and CO₂ sorption sites. Therefore, I rate the level of sophistication of applied solid state NMR experiments as high and intriguing, although I cannot see any conceptual advances or breakthrough findings coming out of these very nice experiments.

The synthetic aspect and discussion of how to tune the defect concentration are interesting but might miss an interesting aspect in the discussion. As rightly stated in the manuscript, formate ligands originate from the reaction of DMF with H₂O to dimethyl amine and formic acid, i.e. formic acid as defect-generating agent / modulator is produced in situ; therewith the amount of formic acid is coupled to the amount of water that is present in DMF. With increasing molar ratios of hydrated metal salt, the water content is increased thereby increasing the amount of formic acid that is available as defect forming agent; an effect that might play a role on top of the changed metal to linker ratio. I understand that the reaction mixture already contains water and that the real molar changes of water due to crystal water are relatively very small; however, I am wondering if the effect of increasing the amount of water by crystal water plays a role? A little bit later in the paper some synthesis experiments of MOF-74 in THF are mentioned; have similar experiments been performed with artificially adding formic acid in THF? These would nicely give a feeling about the sensitivity of MOF-74 for the formation of defects. From the current set of

experiments it seems that the energetics for MOF-74 for formation of defects is relatively small, as "perfect" MOF-74 can be prepared as given in the manuscript. To compare the defect chemistry of MOF-74 with other well-known defective systems would add an interesting point to the overall discussion of the paper.

Together with some other minor suggestions that I have (see below) I conclude that the results are interesting and especially the applied NMR methodology to pin down the spatial arrangement of defects on various length scales might act as a blueprint for future studies on defect MOFs. It is to my knowledge the first paper on MOF-74 where formic acid is used as a modulator for incorporating various amounts of defects, and the results will certainly solidify the communities understanding in the formation of defects in MOFs. As far as I can see, I would also like to add that all experiments including analytics have been performed with care; however, based on the absence of strong conceptual advances that impact the broader field I am recommending rejecting the manuscript. Despite the beautiful solid state NMR experiments, I believe the results find a better home in a more specialized journal.

Minor points:

- * Misuse of word "lattice" on p. 1; see "The lattice sickness pandemic" by Massimo Nespolo
- * In the introduction it is stated "Some studies introduced defects into MOF-74 intentionally by modulating non-native organic ligands¹⁷⁻²⁰"; the main findings as reported in 17-20 should be stated more specifically, so that the novelty of the here applied manuscript is carved out more clearly, i.e. the use of formic acid as modulator to incorporate defects. Also, I'd like to mention at this point that modulation is also the applied synthetic method here, although formic acid as modulator is produced in situ during the synthesis of MOF-74.
- * On p.1 it is given that "However, the existence of intrinsic defects in MOF-74 has only been speculated^{21,22}. Such intrinsic defects are hidden in the conventional X-ray diffraction patterns and could be fundamentally different from those engineered defects". On p.4 it is mentioned that the use of a metal-to-linker ratio of 1:1 delivers the "ideal" MOF-74. Therefore, to speak of hidden / intrinsic defects is misleading in the whole context. In fact, the results in the paper show that by altering the synthetic conditions, intrinsic modulation leads to the generation of defects, while the use of the right conditions delivers the defect free material without any intrinsic / hidden defects.

Reviewer #3 (Remarks to the Author):

This paper reports a work unveiling the presence of formate defects in the well known MOF-74 (particularly the Mg and Ni analogues) and that previous works reported by some of the authors in this same work, missed to report (obviously expressed by the wording "Hidden") and did report the results on MOF-74 analogues (assumed it is purely ordered porous Materials). So many interesting adsorptive properties were reported for different applications (CO₂ capture, CO/H₂ separation, olefin-Paraffin) on a presumably not purely MOF-74 and now this work present an impressive arsenal of characterization data using similar tools (that have been used for other families of MOFs (UIO series) based on Zr. I am quite sceptical that some researchers insist on spending so much effort trying to get defects free Materials..... Most of the commercially used adsorbents/ desiccants are not ordered. In addition scaling up adsorbents will certainly lead to various levels of disorder in the pore network, So my first comment is about the relevance of trying to get defect free MOF in general and MOF-74, in particular while the already reported ones (that seemingly are not defect free) show already unusual results and were published ?

though I am really interested and support heavily the mechanistic studies as shown in this work, I find it is not conveying an unusual message, such conclusion of open metal site alteration vs formate is known. In addition I am uncomfortable that this work has no connection with previous publications on this family of MOFs; For example one of the analogues showed interesting properties for Olefin/paraffin separation (Science 335, 1606-1610 (2012). How this "hidden" defect affected the diffusivity and adsorptive loading performances. And there is many other examples reported by Long and co-workers.

The authors may argue that this is out of the Scope of this specific study but this is not the case because there is a background set on MOF-74 platform that all the results reported before are

associated to the ideal defect free MOF-74.

We are at the stage where MOF is moving to integration in different processes and MOF-74 is one of them. The authors should elaborate in at least one example what will be the economic impact in spending effort of healing defects, at least from the scale-up point of view.

Other scientific-technical comments:

1-At least for one case (CO₂ capture for example) it will be interesting to see the effect of defects on the shape of the heat of adsorption vs CO₂ loading using experiment results to confirm or not the two sites contention suggested by the authors.

2-I disagree with this statement: "the presence of defects is generally established and sometimes even controlled to optimize their Properties 6-9" in fact the healing of defect optimizes the properties not making defects optimize the properties of the MOFs. This is a controversial statement that should be revised.

3- There is no Information about the impact of such discovery on diffusion of molecules. The authors can use the Example of CO₂/CH₄ or C₃H₆/ C₃H₈ example

RESPONSE TO REVIEWERS' COMMENTS

Reviewer #1 (Remarks to the Author):

The manuscript from Fu et al. describes an in-depth investigation of defects in MOF-74 resulting from the decomposition of DMF to formate. The manuscript is succinct and portrays a comprehensive analysis using a combination of multi-dimensional solid-state nuclear magnetic resonance and computational chemistry simulations to discover the effect of ligand substitution with formate and its chemical origin. This is impressive work and outlines how to discover defects for a broad readership, which effectively advertises the transferability of this approach.

I am pleased to find the simulations, used to confirm the experimental results, employed the freely available and established codes cp2k and lammmps. I urge the authors to share the simulation structures, input files and raw data for this study, so that the community is able to reuse these structures identified and build upon this study. For example, models of the defective Mg-MOF-74 structures should be deposited so that other researchers can further investigate these materials.

I would like to commend the authors on this impressive study. The identification of DMF byproducts that form defect sites is an important discovery. This work represents reliable science with a wealth of data and information that is of the quality and novelty expected for publication in Nature Communications.

Response: We thank the reviewer for being very positive on our research. We added the structure cif files and input files for both ideal and defective Mg-MOF-74 as supplementary files.

Reviewer #2 (Remarks to the Author):

Y. Fu and co-workers apply a range of solid state NMR techniques to investigate the concentration, localization, and ramifications of formate-based defects in MOF-74(Mg). The results are complemented by various other experimental techniques such as PXRD, N₂ and CO₂ sorption experiments and MD simulations. The occurrence of defects in MOFs and their impact on the properties, in this case on the gas adsorption mechanism, is an important and likewise challenging area and I rate the general topic as suitable for Nature Communications.

The strong part of the paper is certainly the results coming from the various applied solid state NMR methods. In the most general terms, the results confirm of what can be expected based on chemical intuition. For instance, it is shown that one dobdc⁴⁻ ligand leaves four positive charges unbalanced, and accordingly there is a 4:1 ratio between dobdc⁴⁻ defects and incorporated formate ligands, i.e. the observed stoichiometry is Mg₂(dobdc⁴⁻)_{1-x}(HCO₂⁻)_{4x}. Solid state NMR provides additional spatial information, showing that the substitution pattern as shown in Figure S14 is indeed the most likely scenario. Moreover, the results indicate the absence of any strong correlated defects which have previously been found in other defective MOFs. Lastly, in situ ¹³C NMR of CO₂ adsorbed in the MOF was performed and complemented with MD simulations to show the elimination of open metal sites as the origin for reduced N₂ and CO₂ sorption sites. Therefore, I rate the level of

sophistication of applied solid state NMR experiments as high and intriguing, although I cannot see any conceptual advances or breakthrough findings coming out of these very nice experiments.

Response: We thank the reviewer for being mostly positive on our research. Here, we would like to emphasize that this is the first work which shows the potential existence of formate defects in MOF-74. The findings are crucial to explain the unexpected reduction of CO₂ adsorption (or other properties related to the function of open metal sites) in MOF-74 systems. In addition, we demonstrate comprehensive strategies for people to detect, characterize, and control the formation of formate defects in MOF-74.

The synthetic aspect and discussion of how to tune the defect concentration are interesting but might miss an interesting aspect in the discussion. As rightly stated in the manuscript, formate ligands originate from the reaction of DMF with H₂O to dimethyl amine and formic acid, i.e. formic acid as defect-generating agent / modulator is produced in situ; therewith the amount of formic acid is coupled to the amount of water that is present in DMF. With increasing molar ratios of hydrated metal salt, the water content is increased thereby increasing the amount of formic acid that is available as defect forming agent; an effect that might play a role on top of the changed metal to linker ratio. I understand that the reaction mixture already contains water and that the real molar changes of water due to crystal water are relatively very small; however, I am wondering if the effect of increasing the amount of water by crystal water plays a role?

Response: In fact, water is commonly used in the synthesis and preparations of MOF-74. For example, when 0.2 mmol H₄dobdc is added during synthesis, the precursor solutions of Mg, Co, Zn, and Ni-MOF-74 samples contain variable amount of water (1-6 mL) in addition to the crystalline water. The added water (50-400 mmol) is much greater than the crystalline water associated with metal salt (1-15 mmol). Only a small fraction of water is needed to participate the hydrolysis of DMF. For example, for the Mg-MOF-74 with highest defects, only at most 0.6 mmol water are needed. The amount of crystalline water is not the determinant of defect formation, but the metals themselves (as see in Figure 1d, 1f). The catalytic effect on DMF hydrolysis or the metal-formate binding strength is more relevant for the formation of formate defects. That being said, the mechanism of DMF degradation is not the focus of our study and we would like to leave it for the future research.

A little bit later in the paper some synthesis experiments of MOF-74 in THF are mentioned; have similar experiments been performed with artificially adding formic acid in THF? These would nicely give a feeling about the sensitivity of MOF-74 for the formation of defects. From the current set of experiments it seems that the energetics for MOF-74 for formation of defects is relatively small, as “perfect” MOF-74 can be prepared as given in the manuscript. To compare the defect chemistry of MOF-74 with other well-known defective systems would add an interesting point to the overall discussion of the paper.

Response: Thank the reviewer for the comment. People have actually tried mixing different ligands such as 2-hydroxyterephthalic acid, 2,5-dihydroxybenzoic acid and hydroxyterephthalic acid and produced defective MOF-74 (Chem. Sci. 12, 7324–7333 (2021), Front. Mater. 6, 1–10 (2019)). We

think artificially adding formic acid in THF at appropriate ratio and pH could result in similar defective MOF-74 as well. Because it is an interesting comparison, we would like to leave the artificial control of formate defect to our future ongoing work. We also added a discussion in the last paragraph in text to compare the formate defect in MOF-74 to the well-known correlated defects in UiO-66.

Together with some other minor suggestions that I have (see below) I conclude that the results are interesting and especially the applied NMR methodology to pin down the spatial arrangement of defects on various length scales might act as a blueprint for future studies on defect MOFs. It is to my knowledge the first paper on MOF-74 where formic acid is used as a modulator for incorporating various amounts of defects, and the results will certainly solidify the communities understanding in the formation of defects in MOFs. As far as I can see, I would also like to add that all experiments including analytics have been performed with care; however, based on the absence of strong conceptual advances that impact the broader field I am recommending rejecting the manuscript. Despite the beautiful solid state NMR experiments, I believe the results find a better home in a more specialized journal.

Response: The reviewer missed the key point of our research. This work was not to make some new defects by adding formic acid or by doing special preparation. Instead, we are trying to show that people could introduce formate defects into MOF-74 without notice, because most works used DMF as the solvent. Formate defect could be a commonplace in most MOF-74 materials, and it could have unexpected consequences on their properties (e.g. CO₂ adsorption). To make things clear, we have revised a few sentences in the introduction.

Minor points:

* Misuse of word “lattice” on p. 1; see “The lattice sickness pandemic” by Massimo Nespolo

Response: We changed the word “lattice” to “matrix”.

* In the introduction it is stated “Some studies introduced defects into MOF-74 intentionally by modulating non-native organic ligands¹⁷⁻²⁰”; the main findings as reported in 17-20 should be stated more specifically, so that the novelty of the here applied manuscript is carved out more clearly, i.e. the use of formic acid as modulator to incorporate defects. Also, I’d like to mention at this point that modulation is also the applied synthetic method here, although formic acid as modulator is produced in situ during the synthesis of MOF-74.

Response: Thank the reviewer for the comment. We have added some descriptions of the main findings in references 17-20. Again, the main purpose of our work is to demonstrate the presence of formate defect in MOF74 and to determine its structure. Because the formate defect in MOF-74 was previously unknown and could be neglected by many researchers. On the other hand, our synthetic method indeed provides an easy way to modulate defects in MOF-74.

* On p.1 it is given that “However, the existence of intrinsic defects in MOF-74 has only been speculated^{21,22}. Such intrinsic defects are hidden in the conventional X-ray diffraction patterns and

could be fundamentally different from those engineered defects". On p.4 it is mentioned that the use of a metal-to-linker ratio of 1:1 delivers the "ideal" MOF-74. Therefore, to speak of hidden / intrinsic defects is misleading in the whole context. In fact, the results in the paper show that by altering the synthetic conditions, intrinsic modulation leads to the generation of defects, while the use of the right conditions delivers the defect free material without any intrinsic / hidden defects.

Response: Thank the reviewer for the comment. We called it hidden formate defect because it is not observed in usual XRD characterizations. The defect has not been mentioned by any study previously. But to avoid any confusion, we replace the word "hidden" with "intrinsic" in the title.

We used the phrase "engineered defects" for the defects with intentionally added modulator; the phrase "intrinsic defects" for the defects without additional modulator. We have revised the sentences to make it clear.

Reviewer #3 (Remarks to the Author):

This paper reports a work unveiling the presence of formate defects in the well known MOF-74 (particularly the Mg and Ni analogues) and that previous works reported by some of the authors in this same work, missed to report (obviously expressed by the wording "Hidden") and did report the results on MOF-74 analogues (assumed it is purely ordered porous materials). So many interesting adsorptive properties were reported for different applications (CO₂ capture, CO/H₂ separation, olefin-Paraffin) on a presumably not purely MOF-74 and now this work presents an impressive arsenal of characterization data using similar tools (that have been used for other families of MOFs (UIO series) based on Zr. I am quite skeptical that some researchers insist on spending so much effort trying to get defects free materials.... Most of the commercially used adsorbents/desiccants are not ordered. In addition scaling up adsorbents will certainly lead to various levels of disorder in the pore network, so my first comment is about the relevance of trying to get defect free MOF in general and MOF-74, in particular while the already reported ones (that seemingly are not defect free) show already unusual results and were published?

Response: Indeed, most researchers have not noticed the presence of formate defects in MOF-74 previously. Because the formate defects would not appear in XRD patterns, while most studies relied on XRD as the main characterization technique. We show that NMR characterizations are indispensable, particularly for searching defects in MOFs.

Defects in MOFs have both pros and cons as shown in different studies. But before people can eliminate or take advantage of defects, they must be able to detect, characterize, and quantify defects in MOFs. Our study is the first to show the presence of formate defect in MOF-74 series and we spent great effort to determine the structure. We show that people could introduce formate defects into MOF-74 without notice, because most works used DMF as the solvent. The formate defects could be commonplace in most MOF-74 materials, and it could have unexpected consequences on their properties (e.g. CO₂ adsorption). In addition, the comprehensive SSNMR techniques provide the characterization strategies for people to detect, characterize, and control the formation of formate defects.

though I am really interested and support heavily the mechanistic studies as shown in this work, I find it is not conveying an unusual message, such conclusion of open metal site alteration vs formate is known. In addition I am uncomfortable that this work has no connection with previous publications on this family of MOFs; For example one of the analogues showed interesting properties for Olefin/paraffin separation (Science 335, 1606–1610 (2012)). How this “hidden” defect affected the diffusivity and adsorptive loading performances. And there are many other examples reported by Long and co-workers. The authors may argue that this is out of the Scope of this specific study but this is not the case because there is a background set on MOF-74 platform that all the results reported before are associated to the ideal defect free MOF-74. We are at the stage where MOF is moving to integration in different processes and MOF-74 is one of them. The authors should elaborate in at least one example what will be the economic impact in spending effort of healing defects, at least from the scale-up point of view.

Response: The ability to separate olefin/paraffin is indeed one important application of MOF-74 analogs (Science 335, 1606–1610 (2012)). But as we aware of, CO₂ adsorption is probably the most reported applicational demonstration of MOF-74. This is the very example that we have shown in this work. We showed that the formate defects do have a negative consequence on CO₂ adsorption. The possible strategies to eliminate or heal the defects are to use a proper metal-ligand ratio or to use THF solvent instead of DMF. We think it has been elaborated clear enough in the context. Again, we would like to emphasize that the ability to detect, characterize and control defects is critical to make defect-free or defect-engineered MOFs. That’s why we have spent most of our effort in the characterizations and simulations.

Other scientific-technical comments:

1-At least for one case (CO₂ capture for example) it will be interesting to see the effect of defects on the shape of the heat of adsorption vs CO₂ loading using experiment results to confirm or not the two sites contention suggested by the authors.

Response: Our results show that CO₂ has a higher heat of adsorption in ideal MOF-74 than that in defective MOF-74. This is consistent with the fact that the open metal sites are partially blocked in defective MOF-74. The figure for heat of adsorption (Fig. S7e) is now supplied in supporting information.

2-I disagree with this statement: “the presence of defects is generally established and sometimes even controlled to optimize their Properties 6–9” in fact the healing of defect optimizes the properties not making defects optimize the properties of the MOFs. This is a controversial statement that should be revised.

Response: We have modified this statement in the revised manuscript.

3- There is no Information about the impact of such discovery on diffusion of molecules. The authors can use the Example of CO₂/CH₄ or C₃H₆/ C₃H₈ example

Response: The formate defects could have an impact on the diffusion of molecules, e.g. diffusion

rate and/or diffusion anisotropy. We think it is an important direction to pursue in our future work. But the experimental measurement and detailed analysis of diffusion of small molecules in MOFs is not a simple task (Chem. Mater. 32, 3570–3576 (2020)). It requires dedicated Pulsed-field gradient (PFG) NMR instrument and a laborious effort.

REVIEWERS' COMMENTS

Reviewer #1 (Remarks to the Author):

I commend the authors on including the simulation data as supporting information. I continue to believe this is outstanding and interesting work, appropriate for the readership of Nature Communications.

Reviewer #2 (Remarks to the Author):

Y. Fu have re-submitted their paper „Intrinsic Formate Defects Suppress Adsorption in MOF-74” to Nature Communications, addressing the majority of all previous reviewers’ concerns. After carefully reading the revised version, it is without doubt a well-written, technically very sound, and interesting study which I enjoyed reading. My main concern from my initial assessment, however, “based on the absence of strong conceptual advances that impact the broader field I am recommending rejecting the manuscript” is still there, and I still believe that this study would find a better place in a more specialized journal. The conceptual pendant of this study, i.e., that SSNMR with advanced pulse sequences is suitable for investigating “not only the chemical environment of defects but also their geometries and distributions” (page 1, line 59) was previously reported by the authors. Commenting on “The reviewer missed the key point of our research. This work was not to make some new defects by adding formic acid or by doing special preparation. Instead, we are trying to show that people could introduce formate defects into MOF-74 without notice, because most works used DMF as the solvent.”, I’d like to stress that the main message was/is clearly delivered. I rate the outcomes simply not a surprising finding, although the applied range of techniques, especially the sophistication of SSNMR techniques, is an elegant way of making the case; reviewer three of the initial manuscript shared the same concern (“I find it is not conveying an unusual message”) and the same enthusiasm about the applied techniques.

With leaving the decision about the suitability of the manuscript to the editor, I’d like to raise one last point related to the use of “intrinsic defects” in the title – I am not convinced that this reflects the observed defect chemistry of MOF-74. In semiconductor chemistry and physics, this is a well-established description for point-defects that naturally occur due to thermodynamics. In turn, “intrinsic defects” could relate to any point defects that are observed (or in this case not observed) when synthesizing MOF-74 in THF and occur due to thermodynamic considerations. By turning to DMF as a solvent, the underlying chemistry is changed, i.e., the generation of formic acid, which introduces defects arguably in a non-equilibrium state of the host system, in this case MOF-74; the solid-state chemistry equivalent would be quenching a high-temperature synthesis of an oxide. Therefore, “intrinsic defects” is a bit misleading, although I see why the context might justify the use of “intrinsic”.

Reviewer #3 (Remarks to the Author):

Although I appreciate the effort made by the authors to unveil the Formate based defects but I am find awkward that after so many claims made by the Long and co-workers on the amazing properties of MOF reported and the list is large, now this work teaches us that actually we have hidden formate defects in MOF-74 platform and may need to revisit all those claims. It could be not so Critical for Adsorptive based applications, however my main concerns is about the mixed matrix membrane claims for complex separation C₂H₄/C₂H₆ (Nature materials 15 (8), 845-849) where diffusion is the main driving force.

To make this work stronger, visible and consistent I insist that the authors links these hidden defects to one of the previous works of where diffusion in MOF-74 is dominant (Example of C₂H₄/C₂H₆ separation cited above). The impact or no Impact of this hidden defects on these properties will be an extremely valuable information for the MOF scientific community and can not be disconnected form the scope of this work.

Response to Reviewers' Comments

Reviewer #1 (Remarks to the Author):

I commend the authors on including the simulation data as supporting information. I continue to believe this is outstanding and interesting work, appropriate for the readership of Nature Communications.

Response: We thank the reviewer very much for the support.

Reviewer #2 (Remarks to the Author):

Y. Fu have re-submitted their paper „Intrinsic Formate Defects Suppress Adsorption in MOF-74” to Nature Communications, addressing the majority of all previous reviewers’ concerns. After carefully reading the revised version, it is without doubt a well-written, technically very sound, and interesting study which I enjoyed reading. My main concern from my initial assessment, however, “based on the absence of strong conceptual advances that impact the broader field I am recommending rejecting the manuscript” is still there, and I still believe that this study would find a better place in a more specialized journal. The conceptual pendant of this study, i.e., that SSNMR with advanced pulse sequences is suitable for investigating “not only the chemical environment of defects but also their geometries and distributions” (page 1, line 59) was previously reported by the authors. Commenting on “The reviewer missed the key point of our research. This work was not to make some new defects by adding formic acid or by doing special preparation. Instead, we are trying to show that people could introduce formate defects into MOF-74 without notice, because most works used DMF as the solvent.”, I’d like to stress that the main message was/is clearly delivered. I rate the outcomes simply not a surprising finding, although the applied range of techniques, especially the sophistication of SSNMR techniques, is an elegant way of making the case; reviewer three of the initial manuscript shared the same concern (“I find it is not conveying an unusual message”) and the same enthusiasm about the applied techniques.

Response: We thank the reviewer for appreciating our SSNMR characterizations. Regarding the impact of the findings, we would like to stress that 1) MOF-74 is a benchmark material with lots of applications; 2) The use of DMF is prevalent in MOF synthesis; 3) The presence of formate defects could be much more common than people thought. Knowing the formate defect, many inconsistent results reported in literature regarding the measurement of surface area and adsorption could find answers. Based on our synthetic strategy, people can make better MOFs by minimizing or tuning the density of defects.

With leaving the decision about the suitability of the manuscript to the editor, I’d like to raise one last point related to the use of “intrinsic defects” in the title – I am not convinced that this reflects the observed defect chemistry of MOF-74. In semiconductor chemistry and physics, this is a well-established description for point-defects that naturally occur due to thermodynamics. In turn, “intrinsic defects” could relate to any point defects that are observed (or in this case not observed)

when synthesizing MOF-74 in THF and occur due to thermodynamic considerations. By turning to DMF as a solvent, the underlying chemistry is changed, i.e., the generation of formic acid, which introduces defects arguably in a non-equilibrium state of the host system, in this case MOF-74; the solid-state chemistry equivalent would be quenching a high-temperature synthesis of an oxide. Therefore, “intrinsic defects” is a bit misleading, although I see why the context might justify the use of “intrinsic”.

Response: Thank the reviewer for being rigorous about the term of “intrinsic”. To avoid the ambiguity, we would like to change title into “Solvent-derived defects suppress adsorption in MOF-74”.

Reviewer #3 (Remarks to the Author):

Although I appreciate the effort made by the authors to unveil the Formate based defects but I am find awkward that after so many claims made by the Long and co-workers on the amazing properties of MOF reported and the list is large, now this work teaches us that actually we have hidden formate defects in MOF-74 platform and may need to revisit all those claims. It could be not so Critical for Adsorptive based applications, however my main concerns is about the mixed matrix membrane claims for complex separation C₂H₄/C₂H₆ (Nature materials 15 (8), 845-849) where diffusion is the main driving force.

To make this work stronger, visible and consistent I insist that the authors links these hidden defects to one of the previous works of where diffusion in MOF-74 is dominant (Example of C₂H₄/C₂H₆ separation cited above). The impact or no Impact of this hidden defects on these properties will be an extremely valuable information for the MOF scientific community and can not be disconnected form the scope of this work.

Response: According to established studies (Chem. Mater. 2014, 26, 1, 323–338, Angew. Chem. Int. Ed. 2012, 51, 1857–1860), the adsorption of C₂H₄, C₂H₆ and similar hydrocarbons in MOF-74 is at the open metal sites (see the attached figure), the same for CO₂. So we argue that the representative case of CO₂ is sufficient to clarify the effect of formate defect that it will suppress the adsorption of different gases including CO₂, C₂H₄, and C₂H₆. The further discussion about the selectivity between these gases would be another report with a different purpose. A statement to address this concern is provided in the revised manuscript.

Figure 4. (Top) Structure of acetylene bound to the open Fe²⁺ sites in Fe₂(dobdc), where orange, red, gray, and blue spheres represent Fe, O, C, and D atoms, respectively, and (bottom) the first coordination spheres for the iron centers in the solid-state structures obtained upon dosing methane,¹³⁶ ethane, propane, acetylene, ethylene, and propylene.⁹⁷

Figure extracted from Chem. Mater. 2014, 26, 1, 323–338, which shows the adsorption sites of C₂H₄, C₂H₆ and different hydrocarbons in MOF-74.